# Learning Facts at Scale with Active Reading

**Jessy Lin, Vincent-Pierre Berges, Xilun Chen, Wen-Tau Yih, Gargi Ghosh, Barlas Oğuz**

FAIR at Meta
`jessy_lin@berkeley.edu, {vincentpierre,barlaso}@meta.com`

## Abstract

LLMs are known to store vast amounts of knowledge in their parametric memory. However, learning and recalling facts from this memory is known to be unreliable, depending largely on the prevalence of particular facts in the training data and other factors which are poorly understood. Practitioners are lacking tools which will allow them to ensure that the models learn a given body of knowledge reliably and consistently. To this end, we propose Active Reading: a framework where we train models to *study* a given set of material with self-generated learning strategies. First, we demonstrate models trained with Active Reading on expert domains absorb significantly more knowledge than vanilla finetuning and other data augmentations. We train expert 8B models that achieve 66% on a Wikipedia-grounded subset of SimpleQA (+313% relative over vanilla finetuning) and 26% on FinanceBench (+160% relative over vanilla finetuning) by applying Active Reading to the source documents for each benchmark. Finally, we show that Active Reading can be utilized at pre-training scale to build more factual models. As a demonstration of this, we release WikiExpert-8B, a Wikipedia-expert model trained on 1 trillion generated tokens, which outcompetes models with hundreds of billions of parameters on factual QA.

## 1 Introduction

Large language models (LLMs) have the remarkable ability to store and retrieve a vast amount of world knowledge in their parameters. However, it is poorly understood how to teach models to reliably learn and recall facts from this memory. During pretraining, models struggle to learn the long tail of facts that may appear only sparsely in the pretraining corpora (Kandpal et al., 2023; Wei et al., 2024). Incorporating new knowledge into models during finetuning has also been shown to be brittle, leading to increased hallucinations (Gekhman et al., 2024) or memorization without the ability to manipulate new knowledge (Allen-Zhu & Li, 2024).

While models seem to learn common facts from pretraining, a central challenge in language model training is how to *systematically and scalably* teach LLMs to learn facts without relying on incidental co-occurrence in large-scale web corpora. In this work, we study the question: given a closed body of knowledge, how can we train models to most effectively internalize the information (e.g. to approach perfect factual recall)? Rote memorization of training data is neither sufficient nor desirable for robust knowledge integration. Instead, we aim to promote *generalization*—that is, enabling models to internalize facts in a way that supports accurate reasoning, transfer to novel contexts, and compositional inference.

To that end, we introduce **Active Reading**, a scalable and human-inspired framework that enables LLMs to *study* a given corpus by self-synthesizing data through a diverse set of learning strategies. While models are typically trained with a sequential pass through the training text, humans actively study new knowledge to internalize it. Beyond reading a piece of text once, we might imagine problems to apply a new math theorem to, relate a new fact to what we already know about a person, or understand a concept by explaining it to ourselves in a different way (Brown et al., 2014). Active Reading applies the same principles to synthesize training data for LLMs by allowing the model itself to propose multiple study strategies—e.g., paraphrasing, knowledge linking, active recall, analogical reasoning—and instantiate these strategies on a per-document basis. Unlike existing synthetic data generation methods that rely on fixed templates such as question-answer generation or

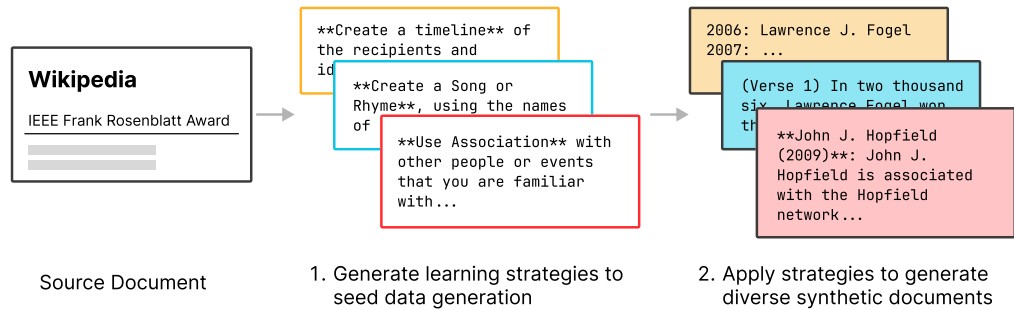

Figure 1: Active Reading as a two-stage synthetic data generation pipeline. In the first stage, the model comes up with diverse learning strategies specific to the given document. In the second stage, strategies are applied independently to generate the self-training data.

simple repetition, this process results in a highly diverse and contextually grounded training signal that emulates the way humans actively engage with new information.

First, we show that Active Reading is highly effective at improving factual recall given a closed body of knowledge (Section 4.1). We achieve state-of-the-art factual recall on SimpleQA by studying the associated Wikipedia documents (Wei et al., 2024), improving factual recall from 16% to 66% (50% absolute, 312% relative) compared to naïvely finetuning on the documents. We also show that Active Reading can be used to train expert models in a specific domain of interest such as finance, improving factual recall by 16% (160% relative) over vanilla finetuning on FinanceBench (Islam et al., 2023).

We find that Active Reading also exhibits improved scaling trends compared to other synthetic data generation strategies as we increase data scale (Figure 2). While paraphrasing and synthetic QA generation plateau in downstream recall performance as we generate more tokens, Active Reading continues to improve as we generate up to 4B words.

Finally, we push Active Reading to pre-training scale by generating 1 trillion tokens of synthetic Wikipedia data to train **WikiExpert**, demonstrating that it can be a viable approach for building more factual base models. WikiExpert (8B) substantially improves factual accuracy on SimpleQA and NaturalQuestions, outperforming substantially larger models including the 236B parameter DeepSeekV2 and 405B parameter Llama 3.1 models and competitive with the state-of-the-art DeepSeekV3 (671B) model.

Taken together, our results suggest that Active Reading is an effective and scalable method to teach models knowledge.

As part of this work, we make the following contributions:

- We propose **Active Reading**, a flexible and scalable pipeline for LLMs to *study* a corpus with self-generated learning strategies, enabling more effective fact learning.

- We demonstrate that studying text with Active Reading leads to substantial improvements (160-312% relative) in factual recall on expert benchmarks and adversarial fact QA tasks, as well as improved scaling behavior as we scale synthetic data generation up to 4B words.

- We scale Active Reading to 1 trillion tokens of synthetic Wikipedia data, demonstrating its viability as a pre-training augmentation. We release WikiExpert, the most factual 8B parameter model to date, and the full 1T-token synthetic dataset to support future research in factual language modeling.

Together, these contributions offer a step towards building reliably factual language models.

## 2 RELATED WORK

**Knowledge Incorporation.** Large pretrained language models (LLMs) have been shown to encode substantial world knowledge, effectively acting as implicit knowledge bases (Petroni et al., 2019; Roberts et al., 2020). Their information storing capacity and ability to use this information scales with model size (Brown et al., 2020; Tirumala et al., 2022; Morris et al., 2025). It has been repeatedly observed that these abilities are inconsistent, and degrade substantially for rare facts (Sun et al., 2024; Wei et al., 2024; Kandpal et al., 2023). Generally, it is not fully understood how to teach models new knowledge after pretraining (Zhao et al., 2025; Ren & Sutherland, 2025; Sun et al., 2025; Lu et al., 2024b; Chang et al., 2024). Simply finetuning on facts, paraphrases, or documents exhibit limited effectiveness or poor scaling, knowledge conflicts (Sun et al., 2025) or unintended side effects like hallucination (Kang et al., 2024; Ghosal et al., 2024). AlKhamissi et al. (2022) presents a comprehensive survey. Our work provides a method to incorporate large amounts of new knowledge at pretraining data scale, and finds that factors such as data diversity and mixing in pretraining data are critical to scale knowledge incorporation.

**Synthetic Data Generation.** Numerous prior works study synthetic data generation for question answering (Du et al., 2017; Alberti et al., 2019; Lewis et al., 2021). These explicitly target factuality with a narrow task focus, however do not consider generalization in the context of general purpose language models. Another line of work, which resulted in the Phi series of models (Gunasekar et al., 2023; Li et al., 2023b; Abdin et al., 2024a;b), investigates the use of synthetic training for LLMs for a wider variety of tasks including language understanding, math and reasoning. However, learning knowledge is an explicit non-goal for this work, and the models do poorly in this area. For knowledge incorporation, several works try direct paraphrasing to improve factual recall, as suggested by Allen-Zhu & Li (2024). Ovadia et al. (2024); Maini et al. (2024); Park et al. (2025) apply this idea at moderate scale. More recently, Yang et al. (2024b) propose synthetic training using the EntiGraph method, which underperformed synthetic QA but showed better generalization. Our method is a superset of these specific data generation methods, and we hypothesize that we outperform them due to increased diversity in generation.

**Domain Adaptation** Domain adaptation of language models for knowledge-intensive expert domains is a large area of literature in itself (Ling et al., 2023), particularly in medical (Peng et al., 2019; Lee et al., 2020; Luo et al., 2022; Singhal et al., 2025), financial (Araci, 2019; Liu et al.; Li et al., 2023a) and legal (Chalkidis et al., 2020; Guha et al., 2023) domains. Most works in this area focus on curating large domain-specific datasets, with only a few focusing on synthetic data generation (Yue et al., 2021). Our method is both general, in that it can be applied to any domain, and specific, in that it naturally adapts to the domain it is applied on, making it unique among prior domain adaptation methods.

## 3 ACTIVE READING

The prevailing method used by practitioners to improve factual coverage of a given knowledge source is simply to increase the mixing weight of that source in the data. However, such passive repetition is known to have limited effectiveness, since too much repetition causes overfitting without generalization. Recent work has established that paraphrasing the data can address this issue to some extent, allowing for more repetition of the target semantic content without overfitting to particular surface forms (Allen-Zhu & Li, 2024). However, paraphrasing is but one strategy for learning. Humans often learn by employing several diverse strategies: ranging from active recall, spaced repetition, asking and answering questions, using diagrams, and many others (Brown et al., 2014).

Some of these strategies might be more effective than others, and may depend on what knowledge we aim to learn. For instance, one might want to study a historical period by putting events on a timeline. Abstract mathematical concepts might be more robustly understood with concrete analogies, or practice problems. Moreover, learning strategies might be complementary, and combining them might be better than using any one of them in isolation. Instead of trying to engineer the best strategy for learning facts, we ask the model itself to come up with a diverse set of learning strategies, given a document we want to study. We then apply each generated strategy to the associated document,

Table 1: Performance on expert domains. SimpleWikiQA tests general tail knowledge, while FinanceBench focuses on domain-specific knowledge. The **info.** subset of FinanceBench consists of questions which pre-dominantly test for information-extraction, while **all** corresponds to the performance on the whole set of questions.

| Model / Method | SimpleWikiQA Model Grader | FinanceBench info. | all |
|---|---|---|---|
| **Llama 3.1 8B Base** | 7.42 | 3.93 | 6.00 |
| repeat (finetune on raw documents) | 15.92 | 18.43 | 10.49 |
| paraphrase | 25.74 | 43.87 | 17.64 |
| synth QA | 47.87 | 44.23 | 17.16 |
| Active Reading (task-agnostic) | 63.33 | 66.18 | 26.83 |
| Active Reading (task-specific) | 66.25 | 61.49 | 25.16 |
| paraphrase + synth QA + Active Reading | 66.66 | 64.45 | 26.12 |
| gold context (Llama 3.1 8B Base) | 65.85 | 84.71 | 44.36 |
| gold context ceiling (Llama 3.1 70B Instruct) | 90.55 | 92.49 | 57.43 |

creating multiple diverse augmentations which can then be used to train the model. We refer to this simple two-stage data generation pipeline as *Active Reading*, which we illustrate in Figure 1.

While one can think of many ways in which to prompt a model to generate *Active Reading* strategies, in this work we instantiate the framework with two prompts. One is a generic prompt which asks the model to generate strategies to study the given material generally, without any particular application in mind (*task agnostic*). The other is a task-specific prompt, where the model is told what downstream task it will be evaluated on (e.g. a trivia competition, or expert financial analysis). This strategy asks the model to first imagine questions from the downstream task, and then come up with strategies that would help it master the type of content asked in its self-synthesized questions. This enables it to generate data that is both (1) more relevant for downstream performance (e.g. focusing on tail facts for a trivia competition), as well as (2) more diverse, by focusing on specific aspects of the document at a time. We refer to this as *task-specific* Active Reading. We provide both prompts in the Appendix D.3.

Regardless of the particular strategy prompt, the resulting learning strategies from *Active Reading* are both diverse and context-specific, providing an advantage over any single fixed strategy. In fact, we find that *Active Reading* recovers many of the previously proposed data augmentation strategies, including paraphrasing (Allen-Zhu & Li, 2024), synthetic question answering (Lewis et al., 2021), and concept maps (Yang et al., 2024b).

## 4 APPLICATIONS

We will evaluate *Active Reading* on two overlapping but distinct use cases.

### 4.1 EXPERT DOMAIN ADAPTATION

First, we investigate whether Active Reading can be an effective strategy to train expert models on a given knowledge-intensive domain, such as medical or finance. In this setting, given a set of documents, we aim to internalize as much knowledge as possible from the documents to solve a downstream task (e.g., question answering). We use the following benchmarks for this purpose:

**FinanceBench** (Islam et al., 2023) is a question answering benchmark grounded on financial disclosure documents of a number of publicly traded companies. The benchmark consists of 10,231 questions, which are split into categories such as numerical reasoning, logical reasoning, and information extraction. Since teaching the model knowledge (rather than reasoning) is the focus of our work, we focus on the "information extraction" category, but also report overall results.

**SimpleWikiQA** is a subset of 3,449 questions from the recent adversarially collected factual question answering benchmark SimpleQA (Wei et al., 2024), where at least one reference document for

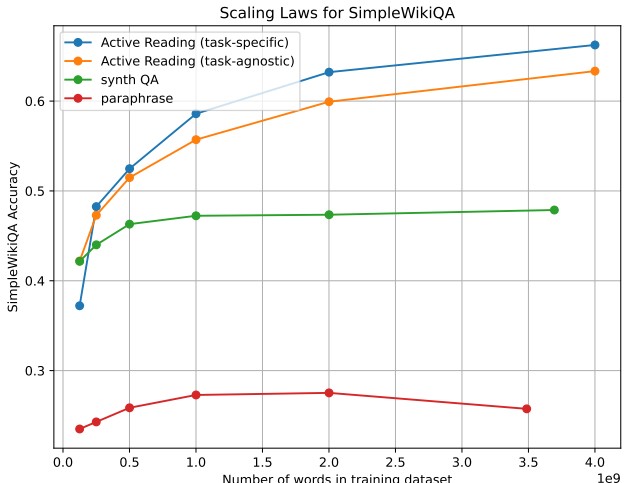

Figure 2: Scaling trends with respect to the number of generated words for each method. While baseline data augmentation strategies like paraphrasing and synthetic QA generation plateau in performance as we scale the amount of synthetic data, Active Reading leads to continued gains in downstream QA accuracy up to 4B generated words.

each question is from Wikipedia. Despite the fact that grounding in Wikipedia ensures that the information has been seen multiple times in the pre-training corpus of every mainstream LLM, the obscure questions in this benchmark remain challenging for current models. In the expert domain setting, we use the set of Wikipedia documents from SimpleWikiQA as our training corpus, evaluating how effectively models can learn tail facts from this small set of documents that have otherwise proven difficult for models to learn.

For baselines, we evaluate training on raw documents with simple repetition, paraphrasing and synthetic question generation. We also provide a baseline where we provide the relevant documents as context to the model (gold setting) in addition to the question, which serves as an upper bound of performance for this task. We do not consider EntiGraph(Yang et al., 2024b), since it was shown to underperform synthetic QA (see their Appendix D therein). For prompts and other implementation details pertaining to baselines, see Appendix D and Appendix A.

For the main experiments in this section, we finetune the Llama3.1 8B base model for 20,000 steps. We fix the number of tokens generated for each baseline to ∼ 4 billion words. During training, we mix in 10% of pre-training data from DCLM (Li et al., 2024) to prevent model degradation. For full training details, see Appendix A.

Results are presented in Table 1. We evaluate the answers of the models using GPT-4o (version *2024-06-01*). Compared to the base model performance (**Llama 3.1 8B Base**), finetuning on the raw documents (**repeat**) provides only small gains in downstream QA performance. Training with data augmentations like **paraphrase** and synthetic question-answer (**synth QA**) provides some improvements. **Active Reading** leads to the strongest factual recall performance on both SimpleWikiQA and FinanceBench, even matching the **gold context** baseline on SimpleWikiQA, where the base model is provided with the documents in context. On FinanceBench, despite good performance on the information extraction subset, there is still a substantial gap with the gold ceiling on the overall benchmark, indicating that on questions that require additional reasoning, in-context processing still outperforms our purely parametric method.

In Figure 2, we show the scaling trends of each method with respect to the size of the generated data up to 4 billion words. This figure confirms that *Active Reading* outperforms other methods across data scales, and maintains a stronger scaling trend, likely due to higher data diversity. In particular, paraphrasing saturates quickly, likely due to the limited number of ways a model can find to paraphrase a particular document. While synth QA demonstrates strong performance at smaller scales, it also plateaus as we scale the number of synthetic tokens.

## 4.2 FACT LEARNING AT SCALE

How does *Active Reading* scale as we increase the size of the corpus of knowledge we want the model to internalize? With a sufficiently scalable approach, one could imagine integrating *Active Reading* into large scale pre-training or mid-training training pipelines, with the hope of mastering

domains with large knowledge corpora (e.g. all medical research) or making base models more factually consistent on a wide set of domains.

To this end, we apply *Active Reading* on the entirety of Wikipedia, generating 1 trillion tokens by augmenting 6 million articles. Training on such a large set of (synthetic) data comes with its own challenges, some of which we go into in the next section.

### 4.2.1 SCALING LAWS FOR FACT LEARNING WITH SYNTHETIC DATA

The SimpleWikiQA reference document set is a subset of Wikipedia, covering only about $0.1\%$ of all documents. When we start expanding this set with more Wikipedia documents in our Active Reading training set, we unsurprisingly see declines in SimpleWikiQA performance. As the pool of documents gets larger, recall of specific facts decrease. This problem is analogous to scaling issues faced by dense (Piktus et al., 2022) and generative (Pradeep et al., 2023) retrieval models. In Figure 3, we expand our training set with 4x and 16x more Wikipedia documents and see that the degradation of performance is substantial, even at this moderate scale (less than 2% of Wikipedia documents).

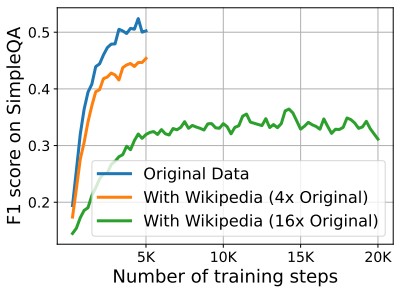

Figure 3: Effect of distractors in a fine-tuning setting. Expanding the set of Wikipedia documents 4x and 16x leads to substantial degradation of performance on SimpleWikiQA.

We find that two major modifications to our training setup greatly improve the scaling behaviour. First, we modify hyper-parameters to more closely resemble (continued) pre-training rather than fine-tuning, by greatly increasing the learning rate (from $1e-5$ to $3e-4$). Second, we increase the weight of pre-training data in our data mix.

First, we hypothesized that with the typical learning rates used for fine-tuning (as in the previous section; Figure 3), the model struggles to allocate enough capacity to learn new facts at scale. Increasing the learning rate forces the model out of its local minima, creating more elasticity for learning. However, the larger learning rate also "breaks" previously learned capabilities, causing decreased performance on guardrail tasks. To encourage the model to recover previous capabilities, we increase the weight of pre-training data in the data mix. In Figure 4, we plot scaling laws as we scale up the data size with these changes. We continue training from Llama-3.1-8B on a data mix of Active Reading-augmented SimpleWikiQA documents (a subset of Wikipedia documents), Active Reading-augmented Wikipedia, and generic pre-training data. We keep the mixing ratio of augmented Wikipedia and pre-training data 1:1 and decrease the relative weight of augmented SimpleWikiQA, so that the relative proportion of SimpleWikiQA data varies between 80% (mostly SimpleWikiQA) to 2.5% (mostly augmented Wikipedia and pre-training data). For the largest scaling run, we do a total of 200k steps with 2.5% augmented SimpleWikiQA (5k steps), 48.75% augmented Wikipedia (97.5k steps), 48.75% pre-training data (97.5k steps). To keep the number of gradient steps we do on SimpleWikiQA constant, we increase the total number of steps as we decrease the relative proportion of SimpleWikiQA. The detailed configurations, details on training dynamics, and performance arc on guardrail metrics can be found in Appendix B.

As seen in Figure 4b, increasing the relative amount of pre-training data recovers guardrail performance on NaturalQuestions(Kwiatkowski et al., 2019), a Wikipedia-based question answering benchmark. Surprisingly, we *also* recover performance on our target task of SimpleWikiQA, even as we scale *down* the relative proportion of augmented SimpleWikiQA training data. The poorly performing intermediate checkpoints still output well-formed and reasonable incorrect answers (e.g. when asked "Who received the IEEE Frank Rosenblatt Award in 2010?", the 40% model responds "geoffrey hinton," an award winner for another year mentioned in the document). This suggests that the typical explanation of catastrophic forgetting of underlying model capabilities does not fully account for the scaling behavior we observe. This poses an interesting question for future work as to why mixing in pre-training data is beneficial for learning, which we discuss in Section 6.

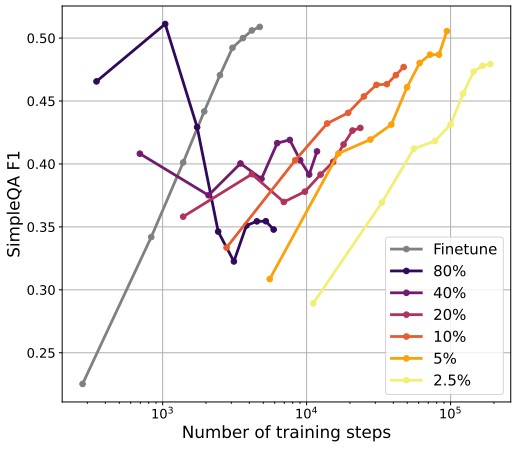
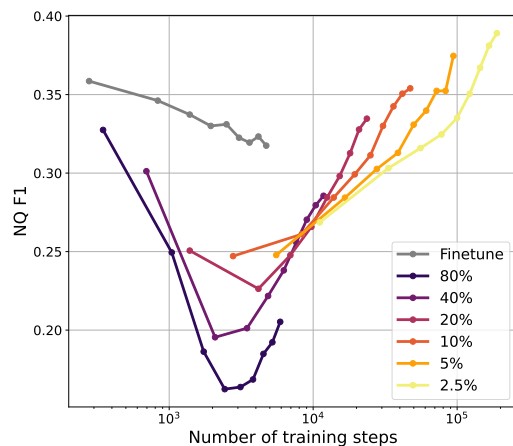

(a) Scaling behavior on SimpleWikiQA.

(b) Scaling behavior on NaturalQuestions (guardrail task).

Figure 4: Continued pre-training as we decrease the proportion of SimpleWikiQA training data from 80% to 2.5% by scaling up the amount of Active Reading Wikipedia and pretraining data. We compare the finetuning baseline (**Finetune** with lr=1e-5, 100% Active Reading SimpleWik-iQA data) against different amounts of Active Reading-augmented SimpleWikiQA, Active Reading-augmented Wikipedia, and pretraining data with a higher learning rate (3e-4). We keep the number of steps on augmented SimpleWikiQA fixed at 5000 but increase the relative amount of augmented Wikipedia and pretraining data, so that the proportion of SimpleWikiQA varies from **80%** to **2.5%**. Surprisingly, when we increase the relative amount of pretraining data, we not only recover guardrail performance, but also recover the original performance on SimpleWikiQA despite doing the same number of steps on this data.

### 4.2.2 TRAINING A WIKIPEDIA EXPERT

Based on these scaling experiments, for our final model we train for 4 epochs over 1T tokens of *Active Reading*-augmented Wikipedia data and 1T tokens of pretraining data, resulting in 8T tokens of training. We do not upweight the SimpleWikiQA documents for this run. We refer to this model as *WikiExpert*, and evaluate it on 3 factuality benchmarks:

- SimpleQA (Wei et al., 2024), and adversarially collected QA benchmark which measures the ability of models to recall tail facts from their pre-training data

- NaturalQuestions (NQ) (Kwiatkowski et al., 2019), a popular Wikipedia-based question answering benchmark which covers a more natural distribution of facts

- TriviaQA (Joshi et al., 2017), a question answering dataset sourced from trivia websites

We summarize results for each benchmark in Table 2. For tail-fact recall as measured by Sim-pleQA, we see that *WikiExpert* improves over it's base model by 222% (7.2 → 23.5), outperforms much larger models including the 236 billion parameter DeepSeekV2 and the 405 billion parame-ter Llama 3.1 models, and is competitive with the state-of-the-art DeepSeekV3 (671B) model. On non-adversarial factual QA benchmarks, *WikiExpert* also improves over its base model (7.6% on NQ and 6.5% on TQA), approaching the performance of a SoTA 72B model. These results suggest that Active Reading is a scalable approach for learning knowledge and has the potential to be useful more broadly in pre-training or mid-training pipelines.

Table 2: Performance comparison across factual QA benchmarks: SimpleQA, Natural Questions (NQ) and TriviaQA (TQA). Qwen and DeepSeek numbers are reported at `https://github.com/deepseek-ai/DeepSeek-V3`.

| Model | SimpleQA | NQ | TQA |
|---|---|---|---|
| Llama 8B | 7.3 | 29.0 | 64.3 |
| WikiExpert-8B | 23.5 | 31.2 | 68.5 |
| Qwen2.5 72B | 9.1 | 33.2 | 71.9 |
| DeepSeekV2 236B | 10.2 | 38.6 | 80.0 |
| Llama 405B | 17.1 | 41.5 | 82.7 |
| DeepSeekV3 671B | 24.9 | 40.0 | 82.9 |

Table 3: Impact of scaling size of the trained model and data-generating model.

| Model | SimpleQA |
|---|---|
| Llama 8B Base | 7.42 |
| with 8B-generated data | 66.25 |
| with 70B-generated data | 62.26 |
| Llama 70B Base | 75.76 |
| with 8B-generated data | 71.52 |
| with 70B-generated data | 77.14 |

## 5 ANALYSIS

### 5.1 UNDERSTANDING ACTIVE READING DATA

What accounts for the difference between data generated by Active Reading and those generated by other synthetic data generation methods—i.e., why is it so much more effective to learn with Active Reading?

First, we investigate whether different methods differ in their *coverage* of the answers in SimpleWikiQA. One hypothesis for why Active Reading is more effective is that generating long-form synthetic documents incorporates knowledge from the source documents more comprehensively, whereas methods like synthetic QA may only help the model to answer a test question correctly if it happens to generate the exact question during data generation. We sample a subset of 100 SimpleWikiQA questions and ask a Llama 3.1 70B Instruct model to judge whether the question is answerable from each synthetic document chunk generated from the source document for that question. In Figure 5, we plot the fraction of answerable questions (i.e. at least one document contains the answer) as we increase the number of synthetic documents, corresponding to training with more synthetic tokens. Synthetic QA data exhibit the highest coverage, despite underperforming Active Reading. Generally, the differences between methods are small, and most questions are answerable after generating 75 document chunks (corresponding to approximately 1M to 2M total generated words on Figure 2). Therefore, we conclude that the differences between the methods are not primarily accounted for by coverage of the target answers.

We hypothesize that Active Reading data is more effective because it is more *diverse*, which may improve knowledge incorporation and learning. We quantify the diversity of the data by measuring Self-BLEU (Zhu et al., 2018), considering each synthetic document as the hypothesis and all other synthetic document as the references and averaging these BLEU scores. Lower self-BLEU scores indicate more diversity (less n-gram overlap) between different synthetic data samples for the same source document. In Figure 6, we compare self-BLEU for 100 sampled source documents as we increase the number of synthetic words generated. As we generate more synthetic tokens, we expect to see self-BLEU scores increase, as it becomes more likely that we generate repetitions of the same way to study the content. Even with few synthetic augmentations per source document, data generated from synthetic QA and paraphrasing is highly self-similar, while data generated by both Active Reading methods is more diverse. Task-specific Active Reading is also more diverse than task-agnostic, aligning with the improved scaling properties that we observe.

### 5.2 SCALING MODEL SIZE

We investigate the impact of scaling model size on performance by additionally training and generating data with 70B models. We show the performance of task-specific Active Reading in Table 3 as we scale model size. Surprisingly, training the 8B model with data generated by the 70B model does not outperform using self-generated data, despite intuitions that 70B-generated data may be

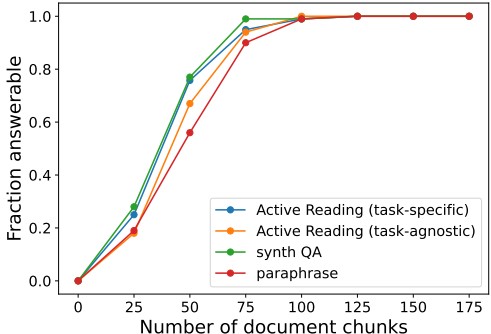 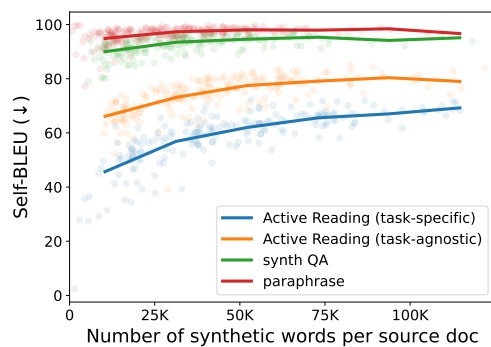

Figure 5: Coverage of n=100 SimpleWikiQA answers in data synthesized by different methods. Most questions are answerable after generating 75 chunks for all methods.

Figure 6: Diversity of data synthesized by different methods, measured with self-BLEU between synthetic data documents for the same source document. We observe that the more effective methods generally exhibit higher data diversity.

more diverse or higher quality. One hypothesis is that training on data that is closer to the model's existing comprehension capabilities and knowledge leads to better learning (i.e. not too challenging or out-of-distribution for the model), and perhaps that it may important to generate the data from the exact model itself. We leave these hypotheses as directions for future work to investigate.

# 6 DISCUSSION & CONCLUSION

**Understanding the role of pretraining data in knowledge acquisition.** In Figure 4, we found that mixing in a large amount of pretraining data was necessary to recover performance on SimpleWikiQA. We kept the number of steps on SimpleWikiQA fixed, decreasing the proportion of this data as we increase the pretraining weight. This poses a question for future work as to why mixing in pre-training data helps with knowledge acquisition: is it because we recover some underlying capability of the base model that is important for task performance, or because some feature of diverse pre-training data induces more "plasticity," making the model more amenable to absorbing or organizing new knowledge (e.g., by reversing knowledge entropy decay (Kim et al., 2024)?). Understanding the properties of pre-training that lead to robust knowledge acquisition may enable practitioners to train models that master any domain of interest, and may be a key milestone towards lifelong learning settings where the model continually acquires new knowledge.

**Parametric knowledge vs. retrieval-augmented generation.** A common approach to improving factuality, particularly on rare domains and tail queries, is retrieval augmented generation (RAG). In this work, we have opted to focus on the closed-book setting. However, the "gold-context" baselines provide a sense of the performance of a RAG system with an oracle retrieval. Modern RAG systems are reported to perform close to this oracle baseline on the benchmarks we evaluated in this work. Therefore our results suggest that there is still a substantial gap between the RAG setting and what the best methods can achieve with parametric updates.

Nevertheless, there are numerous advantages to improving parametric model knowledge (vs. using an external knowledge index). Practically, RAG pipelines are considerably more complex, with larger storage cost, higher memory and compute requirements (due to longer context demanded by additional context) and higher latency, due to the need to query an external index. In the long term, storing knowledge natively in the model's parameters may provide generalization advantages, as the model can relate different pieces of knowledge in its parameters. For example, it is clear a coding model that operates primarily by retrieving similar problems is fundamentally limited in its capabilities. This is also particularly apparent for complex or indirect queries, where simple retrieval augmentation falls short (Yang et al., 2024a). We believe integrating our method with more powerful reasoning models can make progress in these areas.

**Scaling up Active Reading as a training paradigm.** Our scaling results suggest that Active Reading has potential to be scaled up as a standard pre-training or mid-training procedure. One could imagine augmenting training data with Active Reading by default— ensuring the model always "studies" new information in diverse enough contexts for learning, and providing a solution to concerns that pre-training is increasingly limited by data scale (Villalobos et al., 2024).

What would be needed to get to 100% recall on all facts (that we care about) in the pre-training data? Whether or not LLMs can eventually replace search engines seems to be a recurring question (Tay et al., 2022; Bevilacqua et al., 2022). Recent work (Lu et al., 2024a) has suggested that this is an altogether infeasible goal. The capacity of LLMs to store facts with respect to the number of parameters remains an open area of research. Architectures which allow scaling this capacity without increasing FLOPs would be of particular interest in studying these scaling laws (Shazeer et al., 2017; Berges et al., 2024).

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

## A  TRAINING DETAILS

The training runs from Table 1 and Figure 2 had the following hyperparameters: Trained for 20,000 steps, total batch size of 128 and sequence length of 4,096 with a constant learning rate of $10^{-5}$. We report the performance of the best checkpoint after 20,000 steps, evaluating every 2,000 steps.

The training data for the `repeat` method is the raw documents. For the scaling experiments, we generated approximately 4 billion words for each method. The actual word count was: 3,486,858,220 for paraphrasing, 3,694,833,139 for synthetic QA, 4,000,262,090 for active reading task-agnostic and 4,000,330,802 for active reading task-specific. For Figure 2, each dataset was

subsampled from 4B generated words to the indicated number of words. In all runs, we mix in 10% of pre-training data from DCLM (Li et al., 2024), although we did not find it strictly necessary to prevent model degradation at the scale of these experiments and performance improves on SimpleWikiQA by a couple of percentage points if we do not mix pre-training data.

## B  SCALING LAW TRAINING CONFIGURATIONS

For Figure 4, we use the following hyperparameters for each method. We vary the proportions of Active Reading-augmented SimpleWikiQA documents, Active Reading-augmented Wikipedia documents, and generic pretraining data by varying the mixing weights. For all methods, we use a sequence length of 4096 and batch size of 4,194,304 tokens.

Table 4: Hyperparameters for scaling law experiments.

|  | Finetune | 80% | 40% | 20% | 10% | 5% | 2.5% |
|---|---|---|---|---|---|---|---|
| Learning Rate | 1e-5 | 3e-4 | 3e-4 | 3e-4 | 3e-4 | 3e-4 | 3e-4 |
| Weight AR SimpleWikiQA | 100 | 755.2 | 125.87 | 47.20 | 20.98 | 9.94 | 4.84 |
| Weight AR Wikipedia | 0 | 94.4 | 94.4 | 94.4 | 94.4 | 94.4 | 94.4 |
| Weight Pretraining | 0 | 94.4 | 94.4 | 94.4 | 94.4 | 94.4 | 94.4 |
| % AR SimpleWikiQA | 100% | 80% | 40% | 20% | 10% | 5% | 2.50% |
| % AR Wikipedia | 0 | 10% | 30% | 40% | 45% | 47.50% | 48.75% |
| % Pretraining | 0 | 10% | 30% | 40% | 45% | 47.50% | 48.75% |
| # Steps on SimpleWikiQA | 5000 | 5000 | 5000 | 5000 | 5000 | 5000 | 5000 |
| # Total Steps | 5000 | 6250 | 12500 | 25000 | 50000 | 100000 | 200000 |
| Total Tokens | 2.10E+10 | 2.62E+10 | 5.24E+10 | 1.05E+11 | 2.10E+11 | 4.19E+11 | 8.39E+11 |

## C  PERFORMANCE OF WIKIEXPERT ON GUARDRAIL TASKS

Table 5: Benchmark performance of WikiExpert 8B and Llama3.1 8B on guardrail tasks

| Model | OBQA | PIQA | HellaS. | Winog. | ARC-e | RACE-m | MMLU | BOOLQ | GSM8k | MBPP |
|---|---|---|---|---|---|---|---|---|---|---|
| WikiExpert 8B | 44.4 | 81.3 | 80.5 | 75.1 | 78.4 | 64.7 | 65.5 | 83.1 | 48.6 | 39.4 |
| LlaMA3.1 8B | 45.8 | 81.0 | 80.7 | 74.3 | 81.5 | 65.3 | 66.4 | 83.4 | 54.4 | 48.0 |

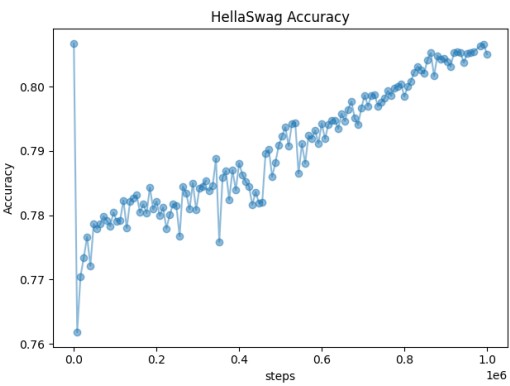

Figure 7: Drop and recovery of performance for guardrail task HellaSwag, mirroring the drop and recovery on NaturalQuestions seen in Figure 4b.

## D  GENERATION PROMPTS

For all generation, {chunk} is replaced by the corresponding document that we are trying to augment.

## D.1 PARAPHRASING

```
Use the information in the following snippet to write an informational
paragraph in your own words.  Make sure to cover all the information,
including all entities, dates and places in the original document.  Do
not add additional material.  Directly output the paragraph and nothing
else, e.g. do not say "Here's the paragraph".

<document>
{chunk}
</document>
```

## D.2 SYNTHETIC QUESTION ANSWER

```
Generate a comprehensive list of all questions and corresponding answers
that can be answered from this document.
Make sure all entities (including people, dates and locations) are
covered.

Output one question-answer pair per line, answer seperated from the
question by a single space.  Questions should be unambiguous, have proper
 capitalization and a question mark. The answers should be as concise as
possible. Do not output anything additional, only output question-answer
pairs (do not start with "Here are the questions and ...").

Example lines of output:
What is the native range of the European fan worm? The northeastern
Atlantic Ocean, the North Sea and the Mediterranean Sea.
What countries are in the European fan worm's native range? The United
Kingdom, Ireland, France, Spain, Portugal, Italy, Greece, and Turkey.
Is the European fan worm found in South America? Yes.
(...)

<document>
{chunk}
</document>
```

## D.3 ACTIVE READING

For active reading, in addition to {chunk} the model is given a strategy. In the following prompt, {strategy} is replaced an active reading strategy. The prompts used to create the strategy are in Section D.3.1 and D.3.2

```
Here's a learning strategy:
{strategy}

Apply this strategy to the following document:
<document>
{chunk}
</document>
```

### D.3.1 TASK AGNOSTIC STRATEGY GENERATION

```
Consider the following document. What are some strategies specific to
this document that I can use to help me learn and remember all of the
information contained? Use markdown and prefix each strategy with ##

<document>
{chunk}
</document>
```

### D.3.2 TASK-SPECIFIC STRATEGY GENERATION

```
I need to study for a trivia competition. Generate a list of questions
that covers every piece of information in this document. After generating
 all the questions, for each question, generate a general study strategy
or prompt that would help me memorize that kind of information (without
focusing too much on the particular question). The prompt should outline
a detailed set of guidelines or step-by-step for how I should rehearse or
 exercise the information to most effectively internalize it.

Output all the questions, then <start_strategies>, then all the
strategies. Prefix each strategy with a ##.

<document>
{chunk}
</document>
```

