# OpenReview forum: "Learning Facts at Scale with Active Reading"
_ICLR.cc/2026/Conference — ICLR 2026 Poster_

### Official Review · Reviewer_2FjT · 2025-10-20

**Soundness:** 2
**Presentation:** 3
**Contribution:** 1
**Rating:** 4
**Confidence:** 4

**Summary:**

The paper propose Active Reading: a two-stage synthetic data pipeline where the model first generates diverse learning strategies for a given source document (e.g., timelines, analogies, Q&A, active recall prompts), and then applies each strategy to produce multiple augmentations used for training.

**Strengths:**

1. The method is clear: first make learning strategies for each document, then apply them to create training data. It covers paraphrase, Q&A, and simple concept linking. It works for general or task-specific goals.
2. The empirical gains are large. On SimpleWikiQA the score reaches about 66 percent. On FinanceBench the score reaches about 26 percent. Both beat repeat, paraphrase, and synthetic-QA under the same token budget.

**Weaknesses:**

1. When the document pool expands beyond the target subset, performance on the target recall task drops materially. The proposed “fix” (higher LR + heavier pretraining-data mix) is effectively a training patch, not a property of the method. If simply adding more relevant-but-broader documents hurts recall, then Active Reading does not positively scale in the straightforward way the paper implies.
2. Most wins of the proposed method are on long-tail factual recall (e.g., SimpleQA subsets). Improvements on broader QA (NQ/TQA) are incremental. There is little evidence that Active Reading scales to domains requiring multi-hop, compositional reasoning, or other forms of tasks like math reasoning or coding.

**Questions:**

1. The paper concedes that modern RAG approaches approach gold-context ceilings on these benchmarks. Given (i) the massive synthetic generation + continued pretraining cost and (ii) the modest gains on non-adversarial QA, why is parameterizing all this knowledge inside weights more efficient than maintaining a high-quality retriever+index?

---

> ### Author Response · Authors · 2025-11-21
>
> Thanks for your review! We're happy to hear that you think the empirical gains shown in our work are compelling.
>
> To address your concerns:
>
> ---
>
> ### Our paper shares learnings on how to make synthetic data generation work
>
> > When the document pool expands beyond the target subset, performance on the target recall task drops materially. The proposed “fix” (higher LR + heavier pretraining-data mix) is effectively a training patch, not a property of the method.
>
> We chose to highlight "things we tried that didn't work" to share our learnings on synthetic data generation. **The ultimate result, demonstrated with WikiLlama, is that Active Reading works if you get the training right.** This does not mean that the synthetic data generation doesn't work, just that the scaling recipe takes some care to configure correctly – as with many methods! – and we wanted to share some naive alternative solutions to make this clear.
>
> **Does our response (and our ultimately positive scaling results, along with WikiLLama) address your concern about whether Active Reading scales?**
>
> ---
>
> ### Active Reading on other domains would be interesting extensions
>
> > There is little evidence that Active Reading scales to domains requiring multi-hop, compositional reasoning, or other forms of tasks like math reasoning or coding.
>
> We acknowledge that we have not tested Active Reading on other types of tasks, and our paper does not make any claims about the method working on reasoning. However, we think achieving these results we did  on factual learning is a prerequisite/foundation for these extensions, which warrant follow-up papers in themselves. We hope that publishing this work would encourage the community to build on this!
>
> ---
>
> ### RAG vs parametric knowledge
>
> > The paper concedes that modern RAG approaches approach gold-context ceilings on these benchmarks. Given (i) the massive synthetic generation + continued pretraining cost and (ii) the modest gains on non-adversarial QA, why is parameterizing all this knowledge inside weights more efficient than maintaining a high-quality retriever+index?
>
> We address this concern in the discussion ("Parametric knowledge vs. retrieval-augmented generation", lines 457-472), and are happy to discuss the main advantages here:
> We view Active Reading as a step towards enabling effective parametric learning more generally. For instance, if we want to teach the model new coding capabilities, or to handle complex or indirect queries, we would expect RAG to fall short (due to the difficulty of retrieval in these settings). While we do not test these applications in our work, we think understanding what works in synthetic data generation (through our empirical experiments and scaling analysis) is a critical step to effective parametric learning.
>
> Practically, RAG pipelines add inference-time complexity and latency (along with additional memory, compute, and storage requirements). "Distilling" knowledge into the model offers a different train-inference compute tradeoff that may be more desirable for some applications.
>
> ---
>
> We have addressed your concerns. **Given our response, would you consider raising your score?**

---

### Official Review · Reviewer_oxz6 · 2025-10-23

**Soundness:** 2
**Presentation:** 2
**Contribution:** 3
**Rating:** 4
**Confidence:** 3

**Summary:**

- It is unclear why not simply input all of the four graph representations together as a baseline. Maybe the result will be better?
- The paper lacks a comparison with human expert performance, even on a subset of the tasks. Including such results would help readers understand how far current LLMs have progressed in graph reasoning relative to human capabilities.
- Table 4 shows that different models prefer different graph representations, yet the training of the Selector depends on a specific model. This implies that a new Selector must be re-trained each time when the model changes, which limits the generality and practicality of the proposed approach.
- The description of the baseline in Section 4.2 is confusing. Does Vanilla prompting mean using the same workflow without additional training, or does it refer to using a single model without the agentic framework? Clarifying this distinction would make the experimental setup easier to understand.

**Strengths:**

- The core idea of this work is both innovative and well-grounded. The proposed Active Reading framework, inspired by human learning behaviors, offers a conceptually sound and intuitively appealing approach to improving factual knowledge acquisition through self-generated data augmentation.

- The work is methodologically solid, with clear experimental design, comprehensive ablation studies, and detailed scaling analyses. The authors provide meaningful comparisons with strong baselines such as paraphrasing and synthetic QA generation, which strengthens the empirical credibility of the findings.

- The scalability and practical relevance of the method are convincingly demonstrated. Applying Active Reading at pretraining scale to develop the WikiExpert-8B model showcases its feasibility for integration into large-scale training pipelines and its potential for building more factual and efficient LLMs.

- The work makes a notable empirical contribution by quantifying factual recall improvements across both general and expert domains. The consistent performance gains and smooth scaling behavior suggest that the proposed approach captures underlying principles of robust knowledge integration rather than dataset-specific effects.

**Weaknesses:**

- In line 240, the authors state that they add mixed pre-training data to prevent model degradation, but they do not provide experiments to demonstrate the occurrence of such degradation.

- The conclusion in lines 274–276 seems rather obvious. Since SimpleWiki can be viewed as representing long-tail knowledge, while the expanded dataset introduces new knowledge, training on new data may naturally interfere with long-tail knowledge retention. This outcome is not surprising.

- Figure 3 is confusing. It is unclear why the curve for the 4× data condition stops at 5k, the same point as the original dataset.

- The conclusion in lines 308–309 is unsurprising, as the model is trained on a related dataset. Recovering performance in this context seems expected and aligns with intuition.

- The experiments in Section 4.1 are conducted only on the LLaMA-3-8B model, which limits the generalizability of the results.

- Lines 242–250 contain several statements that lack proper citations.

- There are a few typos: in line 235, EntiGraph (Yang ...) is missing a space, and in line 308, NaturalQuestions (Kwiat...) is also missing a space.

- In line 288, it is unclear why the learning rate changes from 1e-5 to 3e-4 instead of to 5e-4 or 1e-4. The rationale for this specific choice is not explained.

**Questions:**

See weakness

---

> ### Author Response · Authors · 2025-11-21
>
> Thanks for your review!
>
> **The summary appears to be for a different paper on "graph-related representations" and "agentic frameworks". Is it possible the rating for this paper was entered by mistake?**
>
> We are very happy to hear your feedback that "methodologically solid, with clear experimental design, comprehensive ablation studies, and detailed scaling analyses," "The scalability and practical relevance of the method are convincingly demonstrated." and "the work makes a notable empirical contribution". **We believe these features of the paper suggest a stronger accept. Could you please review our responses and let us know if you would consider raising your score?**
>
> We respond to your concerns below:
>
> ---
>
> ### Effect of mixing in pretraining data
>
> > the authors state that they add mixed pre-training data to prevent model degradation, but they do not provide experiments to demonstrate the occurrence of such degradation.
>
> To clarify, we found that mixing in pretraining data was important in the pretraining-scale experiments (i.e. in experiments in Sec 4.2: Fact Learning at Scale, Fig 4), and chose to mix in some pretraining data for our smaller scale experiments for consistency. However, mixing in pretraining data is not strictly necessary to prevent model degradation at the billion-token scale (i.e. for experiments in Section 4.1: Expert Domain Adaptation). We mention this in Appendix A, but will make this clearer in the main paper!
>
> In Appendix A:
> > In all runs, we mix in 10% of pre-training data from DCLM (Li et al., 2024), although we did not find it strictly necessary to prevent model degradation at the scale of these experiments and performance improves on SimpleWikiQA by a couple of percentage points if we do not mix pre-training data."
>
> ---
>
> ### Generalizability beyond llama models
>
> > The experiments in Section 4.1 are conducted only on the LLaMA-3-8B model, which limits the generalizability of the results.
>
> We acknowledge this, but unfortunately due to the compute intensity of these experiments, we cannot run substantially more experiments. We hope the community can take these positive results and investigate other model families and tasks!
>
> ---
>
> ### Minor: Clarifications
>
> > The conclusion in lines 274–276 seems rather obvious.
>
> Yes, we agree this is unsurprising and emphasize this on lines 274-276: "When we start expanding this set with more Wikipedia documents in our Active Reading training set, we unsurprisingly see declines in SimpleWikiQA performance"
>
> > Figure 3 is confusing. It is unclear why the curve for the 4× data condition stops at 5k, the same point as the original dataset.
>
> We stopped the run (given limited compute) as the trend is already clear: performance degrades as we add more documents.
>
> > The conclusion in lines 308–309 is unsurprising, as the model is trained on a related dataset. Recovering performance in this context seems expected and aligns with intuition.
>
> The sentence in lines 308-309 is "Surprisingly, we also recover performance on our target task of SimpleWikiQA, even as we scale down the relative proportion of augmented SimpleWikiQA training data." **We want to emphasize that this _is surprising_,** given the result in Figure 3. In Figure 3, the model _is_ trained on a "related dataset," but naively adding more augmented documents _does not work_ -- it decreases performance on the target task, motivating our modifications to the training setup. It is only with these modifications that we recover performance. **This is a key finding of our scaling experiments, so if there is any remaining confusion here please let us know if we can help clarify this further.**
>
> > In line 288, it is unclear why the learning rate changes from 1e-5 to 3e-4 instead of to 5e-4 or 1e-4. The rationale for this specific choice is not explained.
>
> We chose this empirically by running ablations on a smaller set of data. We will mention this in the revised paper.
>
> > Lines 242–250 contain several statements that lack proper citations.
>
> > There are a few typos: in line 235, EntiGraph (Yang ...) is missing a space, and in line 308, NaturalQuestions (Kwiat...) is also missing a space.
>
> Thanks, we will fix this in the final draft!
>
> ---
>
> We have addressed all of your concerns.
>
>  **Given that most of these concerns are clarifications and do not seem to take issue with fundamental content of the paper — and the reviewer's acknowledgment that the paper is "methodologically solid, with clear experimental design" and "makes a notable empirical contribution," would you consider raising your score**?

---

> > ### Comment · Reviewer_oxz6 · 2025-11-23
> > **Thank you for the response.**
> >
> > Thank you for the response. They are helpful and I raise my score 4->6.

---

### Official Review · Reviewer_YUiv · 2025-11-04

**Soundness:** 3
**Presentation:** 3
**Contribution:** 3
**Rating:** 6
**Confidence:** 3

**Summary:**

This paper introduces Active Reading, a novel framework designed to improve the factual reliability of large language models. The problem is that LLMs often struggle to learn and recall facts, especially from the long tail of their training data. The proposed method has a model "study" a given corpus by first generating a diverse set of learning strategies specific to a document and then applying those strategies to create varied synthetic training data. This process is inspired by human learning techniques. The authors demonstrate that this method substantially improves factual recall on expert domain benchmarks, showing a 313% relative gain on a subset of SimpleQA and a 160% relative gain on FinanceBench compared to standard finetuning. They also scale this approach to create WikiExpert 8B, a model trained on 1 trillion synthetic tokens that surpasses the factual accuracy of much larger models like DeepSeekV2 and Llama 3.1 405B.

**Strengths:**

1. The Active Reading method is intuitive, scalable, and presents a clever way to generate highly diverse synthetic data by leveraging the model's own capabilities.
2. The empirical results are extremely strong, particularly the performance of the 8B model on Simple WikiQA which nearly matches the gold context baseline.
3. The release of WikiExpert 8B is a significant contribution, as it achieves state of the art factual recall for its size class and provides a powerful, compact model for fact intensive tasks.
4. The scaling analysis in Section 4.2 is very insightful. The discovery that mixing in general pretraining data and using a higher learning rate is necessary to prevent degradation when scaling the knowledge corpus is a valuable finding for the community.

**Weaknesses:**

1. While the method excels at information extraction, its performance on the full FinanceBench benchmark is notably weaker than the synthetic QA baseline. This suggests the generated strategies may not adequately cover complex reasoning, a point the paper acknowledges but does not fully resolve.
2. The finding in Table 3 that data generated by a 70B model leads to worse performance for an 8B model than its own self generated data is highly counterintuitive. This result is not deeply investigated and raises more questions than it answers.
3. The evaluation is heavily focused on Wikipedia based corpora (SimpleQA, NQ, and the main WikiExpert model). While FinanceBench provides one alternative, demonstrating this method's effectiveness in another distinct domain like medicine or law would strengthen the claims of generality.

**Questions:**

See Weakness.

---

> ### Author Response · Authors · 2025-11-21
>
> Thanks for your review!
>
> **We appreciate you highlighting that "the empirical results are extremely strong," "the release of WikiExpert 8B is a significant contribution" and "the scaling analysis is very insightful…a valuable finding for the community."**
>
> Given these notes, we'd like to ask **is there anything else that would lead you to recommend this paper as a stronger accept?**
>
> In particular, you described the following as "Weaknesses" of the paper:
> 1. How does Active Reading extend to reasoning-intensive domains?
> 2. What explains the fact that self-generated data is better than data generated by another model?
> 3. Is Active Reading effective in other domains like medicine and law?
>
> We think (1) and (2) could be interesting standalone papers in themselves, for instance along the lines of recent work like [1] (which investigates how similar synthetic data strategies extend to reasoning) or [2,3] (which investigates the impact of "on-policy data" on learning dynamics; similar studies could be done for learning).
>
> For (3), we were indeed interested in trying domains like medicine and law, but the main challenge we ran into was finding datasets that were not already saturated by current models. Remaining datasets with headroom for improvement are mostly reasoning-intensive. Our work validates Active Reading on factual learning, which we believe is a prerequisite to these more sophisticated tasks, and builds the foundation for others to explore these more difficult domains.
>
> We believe the results in the paper already warrant publication so that the community can build on this method and explore these suggestions, which we also believe are exciting!
> However, our paper already covers a lot of ground with new empirical results and scientific analysis — we think of these more as extensions rather than weaknesses of the current paper.
>
> Given this, would you be willing to reconsider your score?
>
> ---
>
> [1] Ruan et al. 2025, Reasoning to Learn from Latent Thoughts.
>
> [2] Chen et al. 2025, Retaining by Doing: The Role of On-Policy Data in Mitigating Forgetting
>
> [3] Shenfeld et al. 2025, RL's Razor: Why Online Reinforcement Learning Forgets Less

---

### Official Review · Reviewer_yGFJ · 2025-11-07

**Soundness:** 3
**Presentation:** 4
**Contribution:** 3
**Rating:** 6
**Confidence:** 4

**Summary:**

This paper presents a simple, effective method for generating corpora for continued pretraining. Their method first involves first prompting the LLM to generate a large set of learning strategies, which specify methods of paraphrasing or summarizing information from a source document. LMs then use such strategies to generate synthetic corpera for pretraining. The authors apply this method to wikipedia and financial analysis documents, demonstrating gains on wikipedia and finance-based QA tasks. The authors then perform extensive analysis, looking into scaling behavior, the effect of training hyperparameters (learning rate, data mix ratios), and the impact of diversity in the synthetically generated corpora.

**Strengths:**

1. The proposed method is straightforward and demonstrates significant gains compared to baseline methods, in particular for the 8B model.

2. The analysis is thorough and examines a variety of questions about their proposed method. Each analysis section provides valuable insights into why their method works and under what settings it does.

**Weaknesses:**

1. The results with the 70B model have substantially smaller improvements when compared to the 8B -- improving accuracy on SimpleQA <2% versus ~60% according to Table 3. Given this dramatic difference, it would be helpful to see the full results from the primary settings and some of the analysis repeated with this model. Likewise, experimenting with another base model (non-llama) would also help substantiate these results.

2. Including more randomly selected samples of both task agnostic and task specific strategies would help with understanding what the method is actually doing.

**Questions:**

1. While performance seems consistent across most guardrail tasks in Table 5, is there any explanation for why performance drops so much on GSM8k and MBPP?

---

> ### Author Response · Authors · 2025-11-21
>
> Thanks for your review!
>
> **We're encouraged by your feedback that "the proposed method demonstrates significant gains," "the analysis is thorough," and "provides valuable insights into why their method works."**
>
> We respond to your concerns and questions below. We hope that these strengths of the work are already sufficient to warrant a strong accept, so the community can build upon this work after publication. **Could you please let us know if any other information would convince you to raise your score?**
>
> ----
>
> ## Results with 70B vs 8B
> > The results with the 70B model have substantially smaller improvements when compared to the 8B -- improving accuracy on SimpleQA <2% versus ~60% according to Table 3.
>
> The 70B model sees smaller improvements because its performance is already quite good.
> Recent work like SimpleQA Verified [1] suggest that we might be close to the learnable ceiling with the 70B model, as the original SimpleQA dataset contains some noisy and ambiguous samples, and additionally "contains a high degree of redundancy with semantically similar or lexically overlapping questions" (SimpleQA Verified removes ~20% of questions due to these reasons, which is already close to the final performance we achieve of 77%, considering also we take the Wikipedia-grounded subset of SimpleQA).
>
> We agree it is important to validate this work with larger models and other model families – unfortunately, we don't have the compute to do so beyond the extensive experiments in the original paper.
>
> We'd hope that publishing this work would enable the community to take it further, as we believe the original experiments already provide a lot of evidence that Active Reading works!
>
> ----
>
> ##  Examples of task-specific and task-specific strategies
>
> > help with understanding what the method is actually doing.
>
> Thanks for the suggestion! We'll update the final draft with more examples, and show some examples below.
> _Our intuition is that the particular strategies are less important than the fact that they seed diverse data generation_, i.e. it not that "generating a poem" is important, but that it allows the model to see information in many different ways to facilitate knowledge incorporation.
>
> In particular, there's room for improvement to make some of these strategies more tailored for language models (e.g. "using a highlighter" doesn't intuitively make sense for an LLM), but we want to emphasize Active Reading works well even without hand-crafting the prompts!
>
> We think validating this and improving further on the synthetic data is an interesting direction for future work, but in any case we hope the examples we provide are helpful for getting a sense of the data.
>
> ----
>
> ## Performance on GSM8K and MBPP
> We think it's possible that mixing in code/reasoning-specific data may be necessary to preserve performance on these specialized domains, but for the purposes of this work we used DCLM. Alternatively, higher ratios of pretraining data might be helpful, but we did not explore this as it was out-of-scope for the current work.
>
> ----
>
>
> [1] Haas et al. 2025. SimpleQA Verified: A Reliable Factuality Benchmark to Measure Parametric Knowledge.

---

> > ### Author Response · Authors · 2025-11-21
> > **Examples of generated strategies**
> >
> > Task agnostic strategies:
> > ```
> > **Read the document in sections**\nBreak down the document into smaller sections and focus on one section at a time. This will help you to process and retain the information more effectively
> >
> > **Identify key terms and definitions**\nMake a list of unfamiliar terms and definitions as you read the document. Look up any unfamiliar terms in a dictionary or online resources to ensure you understand their meanings
> >
> > **Use a highlighter or notation system**\nHighlight or annotate important information, such as key financial data, regulatory requirements, and company information. This will help you to quickly identify and review the most critical information.
> >
> > **Create a concept map or diagram**\nOrganize the information in a visual format, such as a mind map or concept map, to help you see the relationships between different concepts and ideas.
> >
> > **Take notes on the key points and summarize the main ideas in your own words. This will help you to reinforce your understanding and retain the information more effectively.
> > ```
> >
> > Task specific strategies:
> > ```
> > **Understand Regulatory Bodies**\nTo memorize information about regulatory bodies, create a mental map of key government agencies and their roles. Start by identifying the main regulatory bodies in the US, such as the SEC, and then associate each with its specific functions and responsibilities. To exercise this knowledge, create flashcards with key terms and definitions, and quiz yourself by matching the term with its corresponding definition.
> >
> > **Document Types and Purposes**\nTo memorize the types of documents and their purposes, create a hierarchy of document types (e.g., annual reports, quarterly reports, etc.) and their corresponding purposes. Practice by categorizing different documents and explaining their purposes. You can also create a concept map to visualize the relationships between different document types.
> >
> > **Company Information**\nTo memorize company information, start by breaking down the information into categories (e.g., company name, state of incorporation, etc.). Create flashcards with key terms and definitions, and practice recalling the information from memory. You can also use visualization techniques, such as creating a mind map or diagram, to associate the different pieces of information.
> >
> > **Stock Exchanges and Securities**\nTo memorize stock exchanges and securities, create a mental map of the major stock exchanges and the types of securities listed on each. Practice by matching different securities with their corresponding exchanges. You can also use flashcards to associate key terms with their definitions.
> >
> > **Reporting Requirements**\nTo memorize reporting requirements, create a flowchart or diagram that illustrates the different reporting requirements for the Company. Practice by following the flowchart and explaining the different requirements. You can also use flashcards to associate key terms with their definitions.
> > ```

---

### Meta-Review · Area_Chair_tHRN · 2026-01-08

**Summary:**

This paper presents a novel and generally well-received framework for generating highly effective synthetic corpora for continued pretraining to improve the factual reliability and knowledge recall of LLMs. The core idea is to use LLM first generates diverse, human-inspired learning strategies (like paraphrasing, Q&A, and knowledge linking) and then applies them to documents. The empirical results are competitive, particularly for the 8B model, which demonstrates substantial gains (e.g., a relative gain on a SimpleQA subset) and the release of the SOTA factual WikiExpert 8B model is considered a major contribution by the first three reviewers. Reviewers also commend the thorough analysis on scaling, data mixing, and hyperparameters, which provides valuable, generalizable insights.

The main critiques, however, center on the generalizability and robustness of the method at scale. Specifically, the substantially smaller gains on the 70B model, the counter-intuitive performance drop when 70B-generated data is used to train 8B, and the performance degradation on the target task when the source corpus is broadened are all raised as weaknesses that require more investigation or a stronger explanation. Furthermore, the limited evaluation across diverse domains (only Wikipedia and Finance) and the lack of a comparison to RAG baselines on efficiency are noted as limiting the claims of generality. Despite these weaknesses, three of the four reviewers rate the paper as at or above the acceptance threshold (Reviewer yGFJ, Reviewer YUiv and Reviewer oxz6 after rebuttal), suggesting the contribution and the novelty of the core framework outweigh the current limitations.

After reading the rebuttal, only Reviewer oxz6 provides feedback on the authors' responses, but I believe Reviewer yGFJ's concerns have been largely addressed by the authors. Although the authors have not addressed Reviewer YUiv and Reviewer 2FjT's concerns, I do agree with the authors that the investigation required by these two reviewers is beyond the scope of this paper.  Achieving these results on factual learning is a prerequisite/foundation for the proposed extensions, which warrant follow-up papers in themselves.  Therefore, I tend to accept this paper.

**Reviewer Concerns:**

As mentioned above, I think Reviewer oxz6 and Reviewer yGFJ's concerns are largely addressed by the rebuttal. However, the rebuttal does not address the limitation raised by Reviewer YUiv and Reviewer 2FjT.

**Reviewer Scores:**

After reading the rebuttal, I think only Reviewer oxz6 will change his score (he already did, in the comments, he mentioned that he would raise the score from 4 to 6). But the other three reviewers (score 664) will not change their ratings. For the two reviewers who gave 6, it seems to me that the provided responses are not strong enough to convince them to raise the score from 6 to 8. For the reviewer who gave 4, the authors simply clarify and claim that the further investigation required by the reviewer is beyond the scope of this paper, so I dont think this reviewer will raise the score from 4 to 6.

---

### Decision · Program_Chairs · 2026-01-26

Accept (Poster)